# Phenolic Compounds from Wild Plant and *In Vitro* Cultures of *Ageratina pichichensis* and Evaluation of Their Antioxidant Activity

**DOI:** 10.3390/plants12051107

**Published:** 2023-03-01

**Authors:** Elizabeth Alejandra Motolinia-Alcántara, Adrián Marcelo Franco-Vásquez, Antonio Nieto-Camacho, Roberto Arreguín-Espinosa, Mario Rodríguez-Monroy, Francisco Cruz-Sosa, Angelica Román-Guerrero

**Affiliations:** 1Departamento de Biotecnología, Universidad Autónoma Metropolitana-Iztapalapa, Av. Ferrocarril de San Rafael Atlixco 186, Col. Leyes de Reforma 1a. Sección, Alcaldía Iztapalapa, Ciudad de Mexico 09310, Mexico; 2Departamento de Química de Biomacromoléculas, Instituto de Química, Universidad Nacional Autónoma de México, Coyoacán, Ciudad de Mexico 04510, Mexico; 3Laboratorio de Pruebas Biológicas, Instituto de Química, Universidad Nacional Autónoma de México, Coyoacán, Ciudad de Mexico 04510, Mexico; 4Centro de Desarrollo de Productos Bióticos (CEPROBI), Departamento de Biotecnología, Instituto Politécnico Nacional (IPN), Yautepec 62731, Mexico

**Keywords:** *Ageratina pichichensis*, antioxidant activity, callus culture, cell suspension culture, phenolic compounds

## Abstract

*Ageratina pichichensis*, is commonly used in traditional Mexican medicine. *In vitro* cultures were established from wild plant (WP) seeds, obtaining *in vitro* plant (IP), callus culture (CC), and cell suspension culture (CSC) with the objective to determine total phenol content (TPC) and flavonoids (TFC), as well as their antioxidant activity by DPPH, ABTS and TBARS assays, added to the compound’s identification and quantification by HPLC, from methanol extracts obtained by sonication. CC showed significantly higher TPC and TFC than WP and IP, while CSC produced 2.0–2.7 times more TFC than WP, and IP produced only 14.16% TPC and 38.8% TFC compared with WP. There were identified compounds such as epicatechin (EPI), caffeic acid (CfA), and p-coumaric acid (pCA) in *in vitro* cultures that were not found in WP. The quantitative analysis shows gallic acid (GA) as the least abundant compound in samples, whereas CSC produced significantly more EPI and CfA than CC. Despite these results, *in vitro* cultures show lower antioxidant activity than WP, for DPPH and TBARS WP > CSC > CC > IP and ABTS WP > CSC = CC > IP. Overall, *A. pichichensis* WP and *in vitro* cultures produce phenolic compounds with antioxidant activity, especially CC and CSC, which are shown to be a biotechnological alternative for obtaining bioactive compounds.

## 1. Introduction

*Ageratina pichichensis*, commonly known as “axihuitl”, is a herb belonging to the Asteraceae family, endemic to the Americas and native to Mexico. This plant has been used especially in Mexican folk medicine for treating skin infections, wounds, indigestion and gastritis, and is supported by several scientific studies where the extracts from its aerial parts possess antimicrobial, antifungal, anti-inflammatory, anti-ulcer and healing properties [1,2,3,4,5]. Due to these pharmacological activities, clinical trials have been conducted and demonstrated its efficiency in the treatment of onychomycosis, chronic leg ulcers, diabetic foot ulcers, and vulvovaginal candidiasis [6,7,8,9,10,11]. Because the pharmacological potential of any medicinal plants depends on the secondary metabolites they produce, the extracts from the aerial parts of *A. pichichensis* have stated the presence of a wide variety of bioactive compounds including benzofurans, benzochromenes, terpenes and phenolic compounds [12].

One of the most abundant phytochemical groups in plants are the phenolic compounds, produced trough the secondary metabolism in plants with remarkable physiological and morphological importance. Phenolic compounds range from simple molecules, such as phenolic acids, to large molecules such as tannins, and include different classes such as flavonoids, lignans, xanthones, anthraquinones and hydroxycinnamic acids [13,14]. Among the functional activities displayed by the phenolic compounds, the antioxidant properties are the most sought after and highlighted, due to their ability to stabilize and deactivate free radicals, before they attack targets in biological cells, preventing the action of these radical molecules, involved in the development of multiple acute and chronic disorders such as diabetes, atherosclerosis, aging, immunosuppression, and neurodegeneration [15,16].

Obtaining bioactive compounds from wild plants (WP) generally represents less than 1% wt., which leads to an overexploitation, threat or extinction of plant species [17,18]. *In vitro* plant cell culture is an environmentally friendly alternative to increase the production of bioactive compounds as systems that ensure the continuous production of bioactive compounds [19,20]. *In vitro* plant cell cultures include callus culture (CC) and cell suspension culture (CSC), the latter being one of the most widely used, as it has excellent scaling attributes for the production of bioactive compounds [21,22]. *In vitro* cultures of *A. pichichensis* have been obtained which produce compounds with anti-inflammatory activity [23]; however, there are no reports of the production of phenolic compounds and their possible antioxidant activity.

Therefore, the aim of this work is to establish *in vitro* cultures of *A. pichichensis*, such as germinated plant *in vitro* (IP), CC and CSC, initiated and derived from WP, to enhance the production of bioactive compounds, identifying the major compounds extracted from each culture, and evaluating the antioxidant activity by 2,2-diphenylpicrylhydrazyl (DPPH), 2,2′-azino-bis(3-ethylbenzothiazoline-6-sulfonic acid) (ABTS), and thiobarbituric acid reactive substances (TBARS) assays.

## 2. Results and Discussion

### 2.1. Establishment of the In Vitro Cultures of A. pichichensis

The *in vitro* cultures of *A. pichichensis* were obtained from the germination of seeds excised from WP (Figure 1a). The resultant IP (Figure 1b) were 2 months old and used to obtain the callus cultures (CC) from explants of nodal segments (Figure 1c). In turn, they were cultured for two months and those with a friable appearance were disaggregated and incorporated into a liquid culture medium for obtaining the *A. pichichensis* cell suspension culture (CSC; Figure 1d).

For callus induction, the effect of the concentration of plant growth regulators (PGR): auxin (NAA; α-naphthaleneacetic acid) and cytokinin (KIN; kinetin) in the explants from nodal segments on the morphogenetic response is shown in Table 1. The morphogenetic response was evident at day 20 of culture. Of all the treatments tested, only eight displayed a positive response regarding the callus formation, where T3, T4, and T7 showed percentages higher than 50%. From these data, as preliminary information for establishing the better PGR conditions for callus induction, the higher percentage of callus formation was obtained when NAA and KIN were 4.52–6.79 μM and 0.46 μM, respectively, in treatments T3 and T4. In both cases, the callus formed developed friable appearance and light-yellow to beige color (Figure 2). Treatment T7 also displayed the simultaneous production of callus and roots while T3 and T4 showed the production of callus and plants. Several studies have reported that the combination of auxin and cytokinin favors the production of callus [24,25]. Similar results were obtained by Sánchez-Ramos et al. [26] for the establishment CC from leaf explants of *A. pichichensis*.

The treatments T3 and T4 were transferred to Murashige and Skoog (MS) liquid medium using the same concentration of PGR as in the CC (Figure 2A). The establishment of CSC was determined based on the performance of better callus disaggregation. Treatment T4 displayed greater disaggregation and dispersion of the cells that T3 at day seven of culture. Cell disaggregation when CSC processes are looked for is a parameter of great relevance for ensuring the mass transfer of nutrients to each individual cell, ensuring the cell viability, and improving the production of bioactive compounds. In this work, due to the best disaggregation of callus in the liquid medium, treatment T4 was selected for the establishment of CSC, growth kinetics were maintained for 21 days (Figure 2B), reaching a maximum biomass of 8.32 ± 0.22 g/L at 16 days, a growth rate (µ) of 0.1576 days^−1^ and a doubling time (dt) of 4.39 days^−1^. The following was performed, and the phytochemical characterization was performed at 12 and 16 days of culture. Similar results were obtained for Sánchez-Ramos et. al. [23] on CSC of *A. pichichensis* obtained from leaf explants at concentrations of, differing in this study, in which CSCs were obtained from nodal segment explants.

### 2.2. Quantification of Phenolic Compounds

Phenolics and flavonoids are a type of natural compounds present in plants considered of great relevance due to their important medicinal attributes and benefits for humans [27]. Quantification of phenolic compounds (phenols and flavonoids) was conducted on methanolic extracts from different plant tissues and cultures of *A. pichichensis*: aerial parts from WP, IP (2-month-old), CC (20-days-old), and CSC cultures (12-days-old). Regarding the total phenolic content (TPC), the different extracts of *A. pichichensis* yielded TPC values from 14.24 to 122.12 mg gallic acid equivalent/g dried biomass weight (mg_GAE_/g_DW_), where the CC from treatments T3 and T4 showed significantly higher TPC (*p* < 0.05), followed by WP and CSC at 12 days of culture, and the CSC at 16 days of culture, being IP the extract with the lowest content, as shown in Table 2. Differences between the treatments indicate that probably the culture of CC for 20 days may induce higher stress to the culture due to the consumption of substrate compounds, eliciting the major synthesis of secondary metabolites, approaching to that found in the extracts from the wild plant. For the CSC, the time of culture influenced the yielding of TPC, suffering a reduced content when culture went from 12 to 16 days. In the case of IP cultured for 2 months, the extract exhibited the lowest TPC yield; this could be associated to the requirement of the plant to reach its adult state (around 41–100 days for some plants of the Asteraceae family) [28,29], and being grown in adequate and aseptic conditions; the plant is growing in a process directed towards its primary metabolism for the preferential production of biomass, over the production of secondary metabolites. Total flavonoid content (TFC) ranged between 29.72 to 209.93 mg quercetin equivalent/g_DW_ (mg_QCT_/g_DW_), as described in Table 2, and following the same behavior found for TPC. The CC from T3 and CSC at 12 days of culture showed significantly higher TFC yields (*p* < 0.05), followed by CC obtained from T4 > CSC at 16 days of culture and WP > IP. Once again, IP showed the lowest yield for TFC, agreeing to the lowest content of TPC. Even though WP and PI are closely related to the plant of origin, the differences observed in the production of phenolic compounds and flavonoids are associated with the differences in the biotic and abiotic stress conditions to which plants are exposed. These differences in the environment promote the activation of secondary metabolism as an adaptive response to ensure plant survival under stress conditions [30,31]. Similar results have been obtained on the production of phenolic compounds in plants growing under natural conditions versus *in vitro* plants of *Baccharis antioquensis* (Asteraceae) [32], *Tanacetum parthenium* (Asteraceae) [30], *Thalictrum foliolosum* (Ranunculaceae) [33], *Argylia radiata* (Bignoniaceae) [31], and *Passiflora alata Curtis* (Passifloraceae) [34]. On the other hand, *in vitro* plant cell cultures have been employed as a strategy to increase the production of bioactive compounds, ensuring their continuous production [19,20]. In particular, CSC is considered a simple and cost-effective method that offers the possibility for obtaining bioactive compounds on a large scale [22,35]. Moreover, *in vitro* CC extract exhibited significantly higher TPC and TFC (*p* < 0.05) than WP and IP treatments, agreeing with the reported by Castro et al. [19] for in *Byrsonima verbascifolia* (Malpighiaceae), Coimbra et al. [36] for *Pyrostegia venustaa* (Bignoniaceae), and Arciniega-Carreón et al. [37] for *Ibervillea sonorae* (Cucurbitaceae), who achieved higher production of phenolic compounds in the *in vitro* CC.

Based on these results, CSC produced 2.0–2.7 times more TFC than WP, whereas IP (2-months-old) produced only 38.8% TFC of that obtained in WP. Increased production of phenolic compounds in CSC has also been reported by Arciniega-Carreón et al. [37] in cultivation of *Ibervillea sonorae* (Cucurbitaceae), where TPC production was 10–20% higher than in extracts obtained from plant growing under natural conditions. Dary Mendoza et al. [20] reported that TPC and TFC in suspension cell cultures of *Thevetia peruviana* (Apocynaceae) were higher compared to explants obtained from the plant growing under natural conditions and *in vitro* callus culture. The high content of phenolic compounds in *in vitro* cultures may be due to the presence of growth regulators, especially the combination of auxins and cytokinin, which favor the production of bioactive compounds [38,39]. These results indicate that *in vitro* techniques promote and increase the production of phenolic compounds in *A. pichichensis*.

### 2.3. HPLC Analysis

The extracts of *A. pichichensis* and its cultures were analyzed by HPLC. Previous reports about the phytochemical analysis of species belonging to the Asteraceae family inform the presence of phenolic acids, flavonoids, and terpenoids compounds. Therefore, the elucidation of major bioactive compounds in *A. pichichensis* and its cultures was based on the use of different standard compounds where only six were identify: gallic acid (GA), caffeic acid (CfA), p-coumaric acid (pCA), catechin (CAT), rutin (RUT), and epicatechin (EPI) as shown in Figure 3.

Among the identified compounds, the phenolic acid GA (retention time, rt = 4.908 min), and flavonoids CAT (rt = 12.375 min) and RUT (rt = 17.33 min) were found in WP extracts, whereas only the flavonoid EPI (rt = 14.255 min) was identified for IP extracts. On the other hand, extracts obtained from CC-T3 and CC-T4 were similar in the elution profiles, differing on the height of some signals, being higher for CC-T4. In these extracts, three phenolic acids were identified: GA, CfA (rt = 13.858 min), and pCA (rt = 17.741 min), and the flavonoid EPI. The use of CSC as source of bioactive compounds when culture during 12 o 16 days show chromatographic patterns such as those exhibited by CC extracts, differing in the absence of CfA in CSC-T4 12d. These differences could be related to the different metabolic routes and growth stages of the *in vitro* culture, that lead to enhancing or limiting the production of specific metabolites in response to different stress factors. The production of phenolic acids and flavonoids *in vitro* cultures is frequently reported [36,40,41,42,43,44,45]. In this study, GA was found in all extracts, except IP, while the flavonoids EPI and pCA were produced in all the *in vitro* cultures of *A. pichichensis*. According to the quantitative analysis of the phenolic compounds identified in the extracts (Table 3), there are significant differences between the concentration of each compound in the different treatments, excepting EPI in CC-T3 and CC-T4. The compound with the lowest abundance is GA (CC-T4 > CC-T3 > WP > CSC-T4 16d > CSC-12d) and EPI the most abundant (CSC-T4 16d > CSC-12d > CC-T4 = CC-T3 > IP). According to these results, *in vitro* cultures of *A. pichichensis* were not only able to produce compounds that the WP did not, but also CSCs produced more of them (GA and CAT). Finally, it is highlighted that CSCs produce more compounds such as: CAT, CfA, EPI and pCA in contrast to CC. Similar results were obtained by Ali et al. [46] in CC and CSC of *Artemisia absinthium* L. (Asteraceae) and Modarres et al. [42] in *Salvia leriifolia* Benth (Lamiaceae) cultures. The identified compounds in this study have been investigated for their medicinal and biological properties as antioxidants [47,48], anticancer [49], antimicrobial [50] and anti-inflammatory agents [51]. Despite other compounds were found in the chromatographic characterization, further characterization should be done for elucidating the complete outlook of the bioactive compounds synthesized in the cell cultures treatments, identifying the major compounds and their contribution to the functional activities.

### 2.4. Antioxidant Activity

The advantage of using *in vitro* cell culture is mainly justified by the improvement of bioactive molecules production in shorter times, where the antioxidant activity is one of the most important biological functions, the extracts of *A. pichichensis* were characterized by the capacity of their extracted compounds to scavenge or inhibit free radical molecules through ABTS, DPPH, and TBARS assays. Figure 4 shows the antioxidant capacity of the different extracts of *A. pichichensis* against different free radicals when exposed to concentrations between 10 to 1000 μg dried extract (DE)/mL. Trolox was used as control.

For ABTS inhibition, the extract from WP required lower concentration for inhibiting this radical reaching a plateau around 178 µg**_DE_**/mL, but higher than Trolox. For the rest of treatments, when the antioxidant concentration was below 100 µg**_DE_**/mL, the better antioxidant capacity was performed by IP, followed by CC-T3, and the rest of treatments, but above this concentration, the ihibition of ABTS was modified substantially by the CC and CSC treatments, with a significant reduction of antioxidant capacity in the IP extracts. Regarding the scavenging of the DPPH radical, Trolox and WP extract showed the same trend observed for ABTS, but for this radical CSC-T4 12d, CSC-T4 16d, and CC-T3 were grouped with better antioxidant capacity than CC-T4 and IP extracts. In the case of TBARS, Trolox and WP extract required lower concentrations for the inhibition of lipid peroxidation, followed by CSC-T4 12d, CC-T3, CSC-T4 16d, CC-T4, and IP extracts as those with the lesser antioxidant capacity. For all the treatments, the percentage of scavenging effect on the DPPH and ABTS radicals was increased with increasing the concentration of *A. pichichensis* extracts, where the highest percentage of DPPH radical inhibition was obtained in a concentration of 1000 μg**_DE_**/mL, with percentages of 94.32 ± 0.66, 90.14 ± 1.58, 93.14 ± 0.39 and 94.16 ± 0.19% for WP, IP, CC and CSC extracts, respectively. In the case of Trolox, the highest percentage inhibition was achieved at a concentration of 56.25 μg**_DE_**/mL. Table 2 shows the values for the IC_50_ from each extract required for inhibiting the 50% of the initial concentration of free radical.

Because free radicals are known to play a definite role in a wide variety of pathological manifestations. Antioxidants fight against free radicals and protect us from various diseases. They perform their action either by scavenging the reactive oxygen species or protecting the antioxidant defense mechanisms [15]. The different extracts from *A. pichichensis* showed statistical differences (p < 0.05) in antioxidant activity, their DPPH radical scavenging potential was as follows: WP > CSC > CC > IP, corresponding to IC_50_ values in Table 2. Similar results were obtained in the ABTS assay, the highest percent inhibition was obtained at a concentration of 1000 μg**_DE_**/mL with 99.16 ± 1.19, 96.29 ± 9.27, 98.47 ± 0.56 and 94.47 ± 0.56% for WP, IP, CC and CSC extracts, respectively. The ABTS radical scavenging potential was as follows: WP > CS = CC > IP, agreeing to IC_50_ (Table 2). Similar results were reported by Esmaeili et al. [52], when evaluating the antioxidant activity of plant extract growing in natural conditions and *in vitro* cultures of *Trifolium pratense* L. (plant and callus), the antioxidant potential reported was as follows: WP > CC > PI.

Differences observed between the two assays (DPPH and ABTS) could be related to the chemical nature of the bioactive substances contained in each extract, and to the nature of the radical used to measure this property. In the case of the ABTS radical cation (ABTS+), it is reactive against most antioxidants, and is soluble in both aqueous and organic solvents, allowing it to be applied in a wide range of pH and/or ionic strength [53], whereas the DPPH assay is based mainly on the electron transfer reaction, and the interactions between antioxidants-DPPH· radicals are determined by the structural conformation of the antioxidants. Thus, some compounds react very rapidly with DPPH·, reducing the number of DPPH· molecules in correspondence to the number of available hydroxyl groups in the antioxidant compound. Nevertheless, this mechanism seems to be more complex and the observed reactions are slower in most antioxidants [54]. Hence, the extracts methanolic of *A. pichichensis* contain diverse types of phenolic compounds with different polarities and solvent affinities, affecting the scavenging capacity of the crude extracts.

The antioxidant activity of *A. pichichensis* extracts, evaluated by TBARS assay (Figure 4C), also showed that the highest percentage inhibition was obtained at a concentration of 1000 μg**_DE_**/mL, reaching 94.48 ± 2.17, 62.25 ± 3.08, 90.65 ± 4.17 and 94.04 ± 2.45% for WP, IP, CC, and CSC extracts, respectively. According to the IC_50_ (Table 2), the activity of the extracts was as follows WP >CSC>CC>CC>IP. In general, the best results of antioxidant activity were in WP extracts followed by *in vitro* cultures of *A. pichichensis* (CC and CSC), in the three assays evaluated (DPPH, ABTS and TBARS), especially in the inhibition of lipid peroxidation, evaluated by the production of TBARS. It is noteworthy that even when CC-T3 extract displayed better antioxidant activity, its capacity to disaggregate for leading to CSC cultures was significantly lower than CC-T4, therefore, this culture can be considered as an alternative for good antioxidant compounds production when no CSC establishment is required. Regarding the CSC when cultured during 12 or 16 days, no significant differences were observed for DPPH and ABTS, being significantly better for CSC-T4 12d for TBARS assay. Similar results were reported by Costa et al. [55] on water/ethanol extracts of *Lavandula viridis* L’Hér growing under natural conditions and callus culture, which were efficient in inhibiting lipid peroxidation. The results obtained in this study suggest that *A. pichichensis* extracts contain bioactive compounds that are capable of donating hydrogen to a free radical to eliminate potential damage.

### 2.5. Correlation between Phenolic Content and Antioxidant Activity

To elucidate the relationship between phenol and flavonoid content and antioxidant activity, Pearson’s correlation analysis was applied considering the TPC, TFC and IC_50_ obtained in the DPPH, ABTS and TBARS assays (Figure 5). Based on the results obtained, TPC showed a positive correlation (*p* < 0.05) with TFC, which demonstrates a direct correlation, i.e., the higher the phenol content, the higher the flavonoid content. In the case of TPC, the correlation was negative with DPPH (*p* < 0.05), ABTS (*p* < 0.01) and TBARS (*p* < 0.01). As for TFC, the correlation was also negative, however, with lower significance, for DPPH (*p* = 0.05), ABTS (*p* = 0.05) and TBARS (*p* < 0.05). Therefore, it could be concluded that the relationship of phenol and flavonoid content is inversely proportional to the IC_50_, i.e., the higher the content of bioactive compounds, the higher the antioxidant activity, and the lower IC_50_ value.

## 3. Materials and Methods

### 3.1. Plant Material

Wild plant (WP) and seeds of *Ageratina pichinchensis* were collected in Tepoztlán Morelos, Mexico, identified by Biol. Gabriel Flores Franco and deposited at the HUMO Herbarium of the Universidad Autónoma del Estado de Morelos (UAEM), with the voucher number 33913.

### 3.2. In Vitro Cultures of A. pichichensis

#### 3.2.1. Plant

The seeds of *A. pichichinses* were surface disinfected in a solution of commercial detergent for 10 min; passed through a sodium hypochlorite solution at 10% (*v*/*v*) for 15 min and through an ethanol solution at 10% (*v*/*v*) for 30 s; the material was washed three times with sterile water for 5 min and finally inoculated in Murashige and Skoog medium (MS) [56], supplemented with 30 g/L of sucrose, 2 g/L of phytagel. Plants were kept in a growth chamber at 25 ± 2 °C and in photoperiods of 16 h light/8 h darkness; at the end of 2 months of incubation, a proportion was used for extraction (Section 3.3) and the other for callus induction. The resultant plants were coded as IP.

#### 3.2.2. Callus Induction

Calluses (CC) were obtained following the methodology described by Sánchez-Ramos et al. [26], with some modifications in the concentrations of growth regulators and the type of explant used. The nodal segments of IP were cut into pieces and four explants were transferred into jars containing 25 mL of Murashige and Skoog medium (MS) [56], and plant growth regulators (PGRs): auxin (NAA; 2,4-dichlorophenoxyacetic acid) and cytokinin (kinetin, KIN). Different concentrations of NAA (0.0, 0.45, 2.26, 4.52 or 6.79 μM) were combined with KIN (0.0, 0.46, 1.39, or 2.32 μM) and were kept in a growth chamber at 25 ± 2 °C and in photoperiods of 16 h light/8 h darkness. Callus subcultures were carried out every 20 days and the percentage of callus induction was determined.

#### 3.2.3. Cell Suspension Culture

The 20-day-old calluses were used to establish the cell suspension cultures (CSC). Fresh calluses (5.71 ± 0.71 g) were transferred to 250 mL Erlenmeyer flasks containing 50 mL of MS liquid culture medium and PGRs as in the callus cultures. The CSC were kept under stirring (110 rpm) and incubated at 25 ± 2 °C and in photoperiods of 16 h light/8 h darkness. CSC were sub-cultured every 12 days using 200 µm nylon filters (Whatman No. 1) to obtain a homogeneous cell culture, an inoculum of 40 g/L were transferred to 500 mL Erlenmeyer flasks with 100 mL of liquid culture medium [23].

### 3.3. Preparations of Extracts

Fresh biomass of the aerial parts of WP and IP of *A. pichichensis* (Figure 1) was dried in a drying oven at 40 °C. Samples (1 g) were used to obtain the extracts by sonication (BRANSONIC, CPX1800H, Danbury, CT, USA) at 40 kHz frequency with 10 mL of methanol at room temperature (25 °C) during 15 min. Then, the extracts were centrifuged at 6000× *g* for 15 min. The supernatant phase was recovered, and the pellet was used for a second and third extraction under the same conditions. Finally, the supernatants were mixed, and the solvent was evaporated under reduced pressure, the resultant dried extracts were stored at −70 °C in amber vials until further analysis.

### 3.4. Characterization of Extracts

#### 3.4.1. Total Phenolic Content (TPC)

The total phenolic content (TPC) was determined according to the colorimetric method described by Giordano et al. [31] and Singleton et al. [57], with some modifications. Briefly, an aliquot of 200 µL of each extract was mixed with 1 mL of Folin–Ciocalteu reagent 1:10 (*v*/*v*), after 1 min, 0.8 mL of Na_2_CO_3_ 7.5% (*w*/*v*) were mixed for 30 s and kept for 60 min in the dark at room temperature, after the reaction time the absorbance was measured at 765 nm in a spectrophotometer (Genesys 2, Spectronics, Waltham, MA, USA), gallic acid was used to construct a standard curve (0–100 µg/mL), the results were expressed in mg of gallic acid equivalents (GAE)/g of dried biomass weight (DW) from each treatment of *A. pichichensis* evaluated.

#### 3.4.2. Total Flavonoid Content

The total flavonoid content (TFC) was determined by the colorimetric method described by Barreira et al. [58], an aliquot of 250 µL of each extract was mixed with 1250 µL of distilled water and 75 µL of NaNO_2_ 5% (*w/v*), after 5 min 150 µL of AlCl_3_-6H_2_O 5% (*w/v*) was added, incubated for 6 min at room temperature, then, 500 μL of NaOH (1 M) and 275 μL of distilled water were added, the solution was mixed and the absorbance was measured at 510 nm. Quercetin was used to construct a standard curve (0–400 µg/mL), the results were expressed as mg quercetin equivalents (QCT)/g_DW_ from each treatment of *A. pichichensis* evaluated.

### 3.5. High Performance Liquid Chromatography (HPLC)

The identification of compounds was carried out on solutions prepared from plant extracts and *in vitro* cultures at 1000 ppm and standards compounds at 20 ppm. HPLC analysis was performed following the methodology proposed by Ramirez-Lopez et al. [59] with the following modifications, an HPLC equipment (Shimadzu SPD-10A, SpectraLab Scientific Inc., Markham, ON, Canada) with a Zorbax Eclipse XDB C 18 column (4.6 mm × 250 mm, 5μm) was used. The mobile phases A and B were employed as follows: mobile phase A contained 0.1% formic acid in MilliQ water and phase B contained 0.1% formic acid in acetonitrile (HPLC grade). Data acquisition was applied for 45 min with a total run of 60 min. Gradient elution was as follows: 92% A/8% B, t 0 min; 85% A/15% B at 5 min; 40% A/60% B at 45 min; 40% A/60% B at 55 min; and back to initial conditions 92% A/8% B at 60 min a flow rate of 1 mL/min. The extracts were monitored at a wavelength of 280 and 370 nm. Gallic acid (GA), caffeic acid (CfA), p-coumaric acid (pCA), catechin (CAT), rutin (RUT), and epicatechin (EPI), were used as standard compounds.

### 3.6. Antioxidant Capacity Assays

#### 3.6.1. DPPH Radical Scavenging Activity

The radical scavenging capacity by DPPH was determined according to Dominguez et al. [60]. Fifty µL of extract in methanol at different concentrations were added to an ethanolic solution of DPPH (100 µM, 150 µL) in 96 wells microplates. Mixtures were incubated for 30 min at 37 °C in the dark and their absorbances were measured at 515 nm in a microplate reader Synergy HT™ (BioTek Instruments, Winooski, VT, USA); Trolox was used as standard The DPPH radical scavenging activity (%) was calculated as follows:(1)$${\mathrm{DPPH}}_{\mathrm{Scavenging}\;\mathrm{activity}}\;\left(\%\right)=\left[\frac{A_{0}-A_{1}}{A_{0}}\right]\times 100$$
where *A_0_* is the absorbance of the blank sample (without antioxidant) and *A_1_* is the absorbance of the sample containing the extract. Median inhibition concentration (IC_50_) values were calculated from plotted graph of percentage scavenging activity against the concentration of the extracts and denote the concentration of antioxidant required to inhibit 50% of radical and expressed as μg dried extract (DE)/mL.

#### 3.6.2. ABTS Radical Scavenging Activity

The radical scavenging capacity by ABTS was determined according to Re et al. [61]. The ABTS stock solution was prepared by adding 90.3 mg of ABTS salt and 16.1 mg of K_2_S_2_O_8_ in 25 mL of distilled water. Stock solution was stored in the dark for 16 h at room temperature before use. The ABTS^+^ radical solution was diluted with distilled water until reach an absorbance value of 0.70 ± 0.05 at 734 nm. Afterwards, 1000 μL of diluted ABTS+ radical solution was mixed with 10 μL of different concentrations of extracts. The mixture was allowed to react for 10 min, at 30 °C in the dark and their absorbances were measured at 734 nm in a microplate reader Synergy HT™ (BioTek Instruments, Winooski, USA); Trolox was used as standard. The ABTS+ radical scavenging activity (%) was calculated as follows:(2)ABTSScavenging activity+ (%)=A0−A1A0×100
where *A_0_* is the absorbance of the blank sample (without antioxidant) and *A*_1_ is the absorbance of the sample containing the extract. Median inhibition (IC_50_) values were calculated from the plotted graph of percentage scavenging activity against the concentration of the extracts and the concentration denoted of the antioxidant required to inhibit 50% of radical and expressed as μg_DE_/mL.

#### 3.6.3. Determination of TBARS

##### Animals

Adult male Wistar rat (200–250 g) was provide by Instituto de Fisiología Celular, Universidad Nacional Autónoma de México (UNAM). Procedures and care of animals were conducted in conformity with Mexican Official Norm for Animal Care and Handling (NOM-062-ZOO-1999). They were maintained at 23 ± 2 °C on a 12/12 h light–dark cycle with free access to food and water.

##### Rat Brain Homogenate Preparation

Animal euthanasia was carried out avoiding unnecessary pain with CO_2_. The cerebral tissue (whole brain) was rapidly dissected and homogenized in phosphate-buffered saline (PBS) solution (0.2 g of KCl, 0.2 g of KH_2_PO_4_, 8 g of NaCl, and 2.16 g of NaHPO_4_ ·7 H_2_O/l, pH adjusted to 7.4) as reported elsewhere [60,62] to produce a 1/10 (*w*/*v*) homogenate. Homogenate was centrifuged for 10 min at 800× *g*. The supernatant protein content was measured using the Folin–Ciocalteu’s phenol reagent Lowry. [63] and adjusted with PBS at 2.66 mg of protein/mL.

##### Induction of Lipid Peroxidation and Thiobarbituric Acid Reactive Substances (TBARS) Quantification

As an index of lipid peroxidation, TBARS levels were measured using rat brain homogenates according to the method described by Ng et al. [64], with some modifications. Supernatant (375 µL) was added with 50 µL of 20 µM EDTA and 25 µL of each extract concentration solved in methanol (25 µL of methanol for control group) and incubated at 37 °C for 30 min. Lipid peroxidation was started adding 50 µL of freshly solution FeSO_4_ 100 µM and incubated at 37 °C for 1 h. The TBARS content was determined as described by Ohkawa et al. [65], with 500 µL of TBA reagent (0.5% 2-thiobarbituric acid in 0.05 N NaOH and 30% trichloroacetic acid, in 1:1 proportion) added at each tube and cooled on ice for 10 min, centrifugated at 13,400× *g* for 5 min and heated at 80 °C in a water bath for 30 min. After cooling at room temperature, the absorbance of 200 µL of supernatant was measured at 540 nm in a Bio-Tek Microplate Reader Synergy HT. Trolox was used as standard. Concentration of TBARS was calculated by interpolation in a standard curve of tetra-methoxypropane (TMP) as a precursor of malondialdehyde (MDA) [66]. Results were expressed as nmol of TBARS per mg of protein. The percentage of inhibition of lipid peroxidation (IR%) was calculated as follows:(3)$$\mathrm{IR}\;\left(\%\right)=\left[\frac{A_{0}-A_{1}}{A_{0}}\right]\times 100$$
where *A_0_* is the absorbance of the blank sample (without antioxidant) and *A_1_* is the absorbance of the sample containing the extract. Median inhibition (IC_50_) values were calculated from a plotted graph of percentage scavenging activity against the concentration of the extracts and denoted the concentration of antioxidant required to inhibit 50% of radical and expressed as μg_DE_/mL.

### 3.7. Statistical Analysis

All the experiments were carried out in triplicate and the results were expressed as means ± SD. TPC, TFC, and antioxidant activity (DPPH, ABTS and TBARS) were compared using the Tukey and ANOVA test. A Pearson correlation test was performed to establish significant effects among the variables: TPC, TFC, antioxidant activity by DPPH, ABTS and TBARS assays. Significant levels were defined at *p* < 0.05 and *p* < 0.01. Statistical analysis was tested with SigmaPlot 12.5 software (Systat Software Inc, 2011, Palo Alto, CA, USA). A Pearson correlation test was performed in R Core Team (2022): R: a language and environment for statistical computing.–R Foundation for Statistical Computing, Vienn.

## 4. Conclusions

*In vitro* cultures (CC and CSC) were obtained from the IP nodal segment of *A. pichichensis* in the presence of a combination of NAA and KIN in MS medium. The disaggregation of the CC was a decisive factor for the establishment of CSC, because only one treatment exhibited disaggregation in liquid medium. On the other hand, *in vitro* CC and CSC cultures were able to produce compounds that WP did not, and CSC produced more phenolic compounds than CC. Based on these results, *in vitro* CC and CSC cultures are an alternative for obtaining phenolic compounds continuously and in a shorter time than plants growing under natural conditions of *A. pichichensis*. However, although *in vitro* cultures were able to produce a higher number of phenolic compounds, the antioxidant activity obtained in the different assays evaluated were significantly lower than in WP. These differences could be related to the affinity of the extracted compounds for the different radicals evaluated, so, a more robust characterization of *A. pichichensis* extracts is necessary for the identification and quantification of the majority compounds and their contribution in the antioxidant activity. Finally, TPC and TFC showed a correlation inversely proportional to the IC_50_ found in the DPPH, ABTS and TBARS assays. With this information we can say that WP and *in vitro* cultures of *A. pichichensis* can produce phenolics compounds with antioxidant activity and that *in vitro* cultures such as CC and CSC are a powerful biotechnological alternative for obtaining bioactive compounds.

## Figures and Tables

**Figure 1 plants-12-01107-f001:**
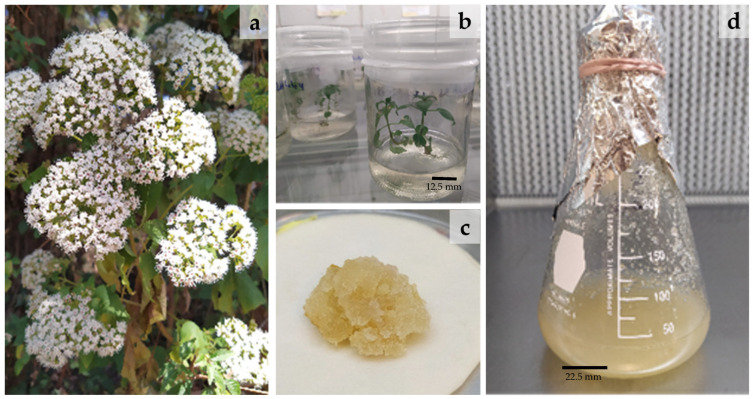
*Ageratina pichichensis*: (**a**) Wild plants (WP) collected at the natural site, (**b**) germinated plant *in vitro* (IP), (**c**) callus cell culture (CC), and (**d**) cell suspension culture (CSC).

**Figure 2 plants-12-01107-f002:**
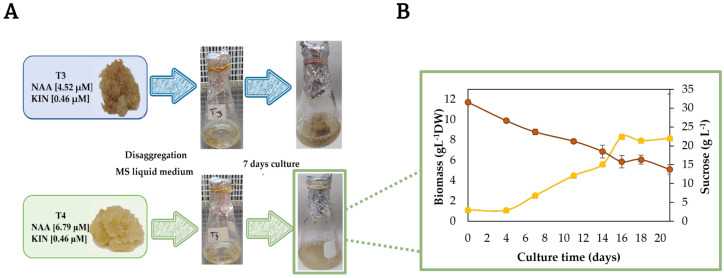
(**A**) Establishment of A. pichichensis cell suspension culture (CSC) in liquid Murashige and Skoog medium (MS) from callus culture (CC) treatments T3 and T4, maximum disaggregation in CSC was achieved at seven-day culture. (**B**) Kinetic growth of T4 suspension cell culture (CSC) during 21 days of culture.

**Figure 3 plants-12-01107-f003:**
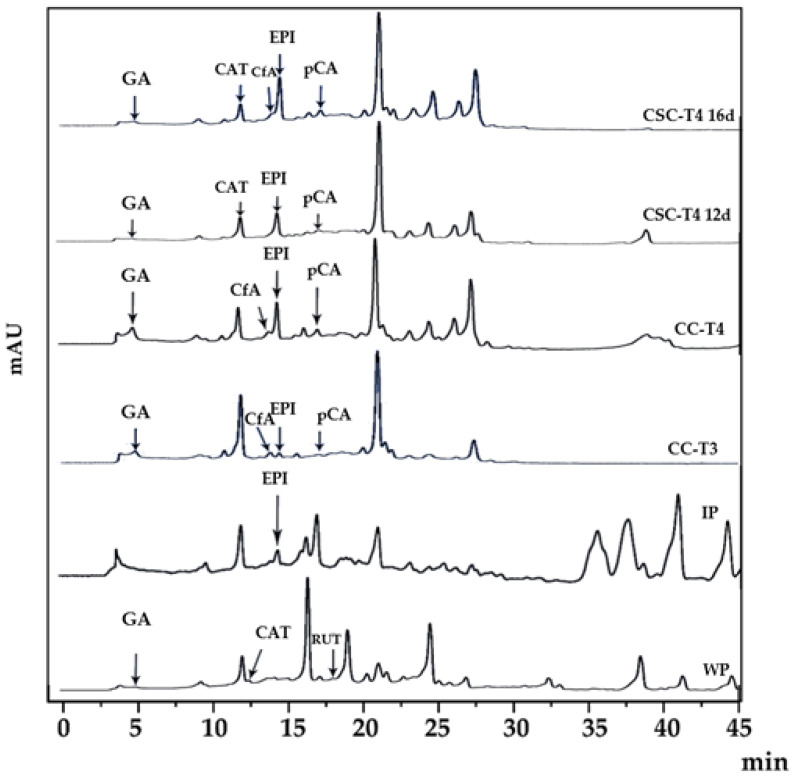
HPLC chromatograms for methanol extracts of *A. pichichensis* from wild plant (WP), *in vitro* plant (IP), callus culture (CC-T3 and CC-T4), suspension cell culture (CSC-T4 12d and CSC-T4-16d). Gallic acid (GA), catechin (CAT), caffeic acid (CfA), epicatechin (EPI), p-coumaric acid (pCA) and rutin (RUT).

**Figure 4 plants-12-01107-f004:**
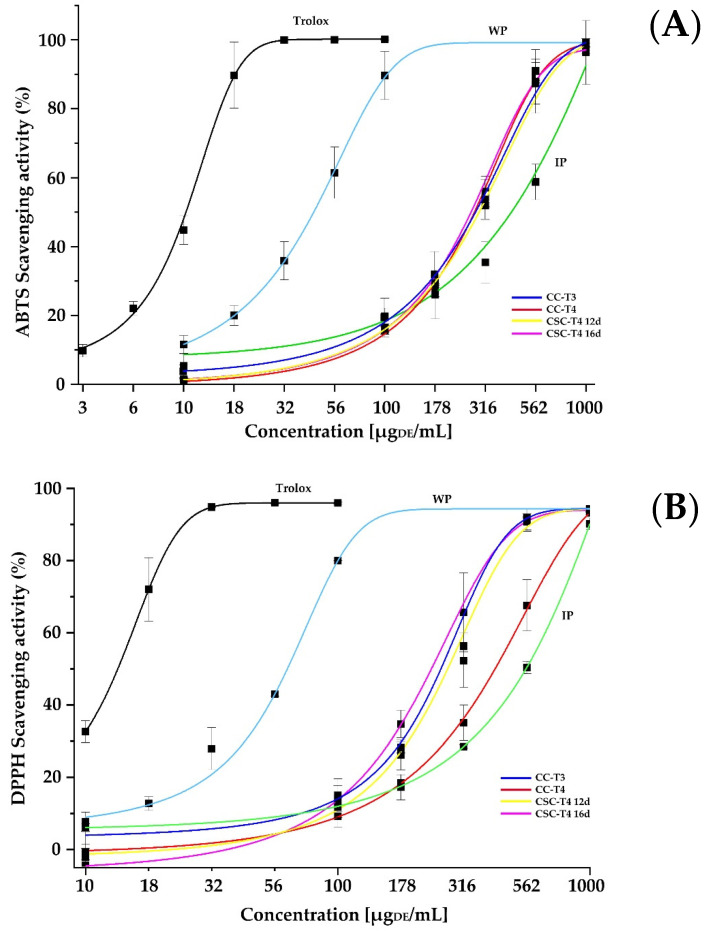
Antioxidant capacity of extracts from *A. pichichensis* and its cell cultures against (**A**) ABTS, (**B**) DPPH radicals and (**C**) TBARS lipid peroxidation.

**Figure 5 plants-12-01107-f005:**
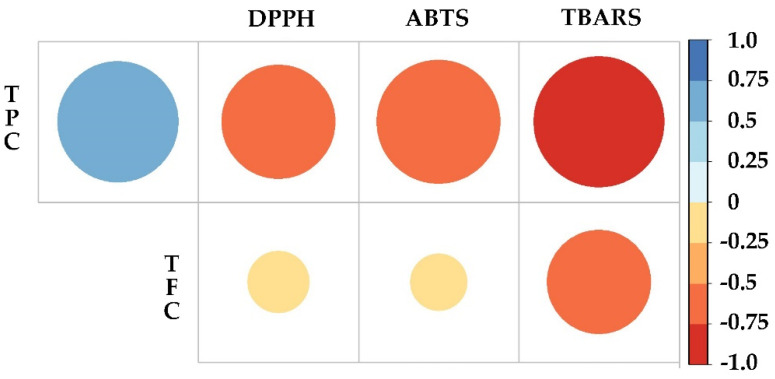
Pearson correlation coefficients for the antioxidant activity characterization. TPC, total phenolic compounds; TFC, total flavonoid compounds; ABTS and DPPH radical scavenging activity, TBARS inhibition of lipid peroxidation.

**Table 1 plants-12-01107-t001:** Effect of plant growth regulators on the morphogenetic response of *A. pichichensis* nodal segment explants at 20 days of culture.

Treatment	PGR		Callus Induction Response			Callus and Roots Induction Response			Callus and Plants Induction Response		
	(μM)		(%)	Fresh Weight (g)	Dried Weight (g)	(%)	Fresh Weight (g)	Dried Weight (g)	(%)	Fresh Weight (g)	Dried Weight (g)
	NAA	KIN									
Control	0	0	ND	ND	ND	ND	ND	ND	ND	ND	ND
T1	0.45	0.46	ND	ND	ND	ND	ND	ND	ND	ND	ND
T2	2.26	0.46	ND	ND	ND	ND	ND	ND	ND	ND	ND
T3	4.52	0.46	93.75 ± 12.50 ^a^	5.25 ± 1.69 ^a^	0.19 ± 0.06 ^a^	ND	ND	ND	37.50 ± 25.00 ^a^	6.72 ± 1.63 ^a^	0.23 ± 0.08 ^a^
T4	6.79	0.46	81.25 ± 12.51 ^a^	6.42 ± 2.00 ^a^	0.24 ± 0.09 ^a^	ND	ND	ND	16.66 ± 14.43 ^b^	16.75 ± 3.17 ^b^	1.85 ± 0.73 ^b^
T5	0.45	1.39	37.50 ± 14.43 ^b^	3.29 ± 0.76 ^b^	2.31 ± 0.3 ^b^	ND	ND	ND	ND	ND	ND
T6	2.26	1.39	18.75 ± 12.50 ^c^	1.46 ± 0.24 ^c^	0.97 ± 0.15 ^c^	ND	ND	ND	ND	ND	ND
T7	4.52	1.39	68.75 ± 12.50 ^b^	1.53 ± 0.18 ^c^	0.70 ± 0.08 ^d^	18.75 ± 12.52	3.83 ± 0.84	1.75 ± 0.83	ND	ND	ND
T8	6.79	1.39	ND	ND	ND	ND	ND	ND	ND	ND	ND
T9	0.45	2.32	ND	ND	ND	ND	ND	ND	ND	ND	ND
T10	2.26	2.32	45.83 ± 29.46 ^b^	4.70 ± 1.43 ^b^	3.00 ± 0.48 ^b^	ND	ND	ND	ND	ND	ND
T11	4.52	2.32	ND	ND	ND	ND	ND	ND	ND	ND	ND
T12	6.79	2.32	ND	ND	ND	ND	ND	ND	ND	ND	ND

ND means Not detected. Different superscripts in the same column mean significant differences at *p* < 0.05.

**Table 2 plants-12-01107-t002:** Phytochemical characterization and antioxidant activity of extracts from *A. pichichensis* and its cell cultures.

Treatment	TPC	TFC	ABTS	DPPH	TBARS
	(mg_GAE_/g_DW_)	(mg_QCT_/g_DW_)	IC_50_ (µg _DE_/mL)		
WP	100.56 ± 6.83 ^bc^	77.26 ± 4.16 ^c^	45.09 ± 6.29 ^a^	60.24 ± 2.66 ^a^	27.85 ± 0.29 ^a^
IP	14.24 ± 1.71 ^e^	29.72 ± 3.78 ^d^	459.68 ± 59.32 ^c^	557.29 ± 16.49 ^d^	783.99 ± 28.06 ^e^
CC-T3	122.12 ± 3.44 ^a^	209.93 ± 0.57 ^a^	284.77 ± 24.52 ^b^	242.58 ± 23.24 ^b^	260.49 ± 9.48 ^c^
CC-T4	113.33 ± 4.46 ^ab^	155.60 ± 12.49 ^b^	282.81 ± 9.01 ^b^	422.67 ± 49.48 ^c^	362.72 ± 12.22 ^d^
CSC-T4 12d	91.89 ± 4.83 ^c^	196.93 ± 2.08 ^a^	282.2 ± 3.28 ^b^	229.96 ±30.81^b^	157.01 ± 5.27 ^b^
CSC-T4 16d	61.47 ± 6.37 ^d^	88.5 ± 9.19 ^c^	285.04 ± 45.12 ^b^	276 ± 15.53 ^b^	234.31 ± 6.52 ^c^

TPC, total phenolic compounds; TFC, total flavonoid compounds; ABTS and DPPH radical scavenging activity, TBARS inhibition of lipid peroxidation. Different letters in the same column mean significant differences between treatments (*p* < 0.05).

**Table 3 plants-12-01107-t003:** Quantitative analysis of identified phenolics compounds of extracts from *A. pichichensis* wild plant (WP), *in vitro* plant (IP), callus culture (CC-T3 and CC-T4), suspension cell culture (CSC-T4 12d and CSC-T4-16d).

Treatment	Identified Phenolic Compounds					
	µg GA/g_DW_	µg CAT/g_DW_	µg CfA/g_DW_	µg EPI/g_DW_	µg pCA/g_DW_	µg RUT/g_DW_
WP	1.75 ^a^ ± 0.06	2.58 ^a^ ± 0.08	Ni	Ni	Ni	3.36 ± 2.20
IP	Ni	Ni	Ni	34.24 ^a^ ± 6.18	Ni	Ni
CC-T3	2.46 ^b^ ± 0.04	Ni	7.14 ^a^ ± 0.28	97.91 ^b^ ± 8.73	3.53 ^a^ ± 0.02	Ni
CC-T4	3.92 ^c^ ± 0.09	Ni	8.11 ^b^ ± 0.03	91.17 ^b^ ± 4.99	1.53 ^b^ ± 0.14	Ni
CSC-T4 12d	0.57 ^d^ ± 0.04	9.97 ^b^ ± 1.33	Ni	145.10 ^c^ ± 3.36	2.45 ^c^ ± 0.07	Ni
CSC-T4 16d	1.17 ^e^ ± 0.04	6.56 ^c^ ± 1.16	14.73 ^c^ ± 0.51	195.87 ^d^ ± 3.77	5.06 ^d^ ± 0.31	Ni

Gallic acid (GA), catechin (CAT), caffeic acid (CfA), epicatechin (EPI), p-coumaric acid (pCA), rutin (RUT). Ni = Not identified. Different letters in the same column mean significant differences between treatments (*p* < 0.05).

## Data Availability

No new data were created or analyzed in this study. Data sharing is not applicable to this article.

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
