# Peer review of "Phenolic Compounds from Wild Plant and In Vitro Cultures of Ageratina pichichensis and Evaluation of Their Antioxidant Activity"

_plants, 2023, doi:10.3390/plants12051107_

Round 1

Author Response

Response to reviewers

The authors thank the comments and suggestions made by the reviewers, they help us to improve our manuscript, the response to each query is given below. All the changes were marked in red in the revised version of the manuscript.

Reviewer 1

Q1. The plant cell culture technique is a useful technique for plant roots or shoots production. It is interesting and important that this technology is used to produce bioactive compounds (phytochemicals) in plants. However, whether the secondary metabolites produced by plant cell culture have the same specific components and biological activities as the wild adult plant is worth exploring. Although the author has identified different phytochemicals in NP and other cell culture strain (IP, CC, and, CSC), the important components that contribute to the antioxidant activity is still unclearly. Some comments and suggestions are presented below to improve quality and to clarify some information.

R. We appreciate the comments from Reviewer 1.

Q2. Some abbreviations of noun have to be revised. Does the “NP” mean wild plant (on page 4, line 120) or adult plant specimen? N means…? and P means…?

R. To better understanding, the meaning of the adult wild plant, the term NP has been replaced with WP, which means Wild Plant.

Q3. On page 13, line 353, DW means “dry weight of extracts”.

R. DW corresponds to the dried weight of biomass used for obtaining each extract (WP, IP, CC-T3, CC-T4, CSC-T4). This information was explicitly defined in the manuscript.

Q4. It is inappropriate to use the abbreviation “TEAC” as Trolox equivalent. It can be revised to TE.

R. The nomenclature and units for reporting the antioxidant activity in the extracts were revised and corrected.

Q5. On page 14, line 389, 406, and 446, IC50 means “half maximal inhibitory concentration”, not “median inhibitory concentration.”

R. The term IC50 was defined in the manuscript as the concentration of an antioxidant-containing substance required to scavenge 50% of the initial radical (Mishra et al., 2021, Alonso-Carrillo et al.,2017).

Mishra et al., 2021. DOI: https://doi.org/10.1007/s11696-021-01708-6

Alonso-Carrillo et al., 2017 DOI: http://dx.doi.org/10.1016/j.indcrop.2017.04.002

Q6. On page 15, line 441, the definition of IR is unclearly.

R. The definition of the term “IR” was improved for making it clearer in the manuscript.

Q7. Please showed correct significant figures of data in Table 1.

R. All the significant figures for data in Table 1 were revised.

Q8.  In Table 1, Why does some data show excessively high standard deviation? Too high standard deviations of data mean poor repeatability, and it will reduce confidence.

R. The process of plant cell induction to callogenesis culture is a preliminary step for obtaining and defining the conditions for achieving cell suspension cultures, in this step the main objective is to establish the effect of the plant growth regulators and their concentrations on the development of differentiated or dedifferentiated cells, depending on the objective pursued. Most of the works reported in the literature, despite stating the use of replicates, report the callogenesis yield only as the mean value (Liu et al., 2021; El-Shafey et al., 2019; Jamil et al.,2018; Gao et al., 2011 and Siwach et al., 2011), without considering the standard deviation (SD). Moreover, some other research that includes SD values (Deng et al., 2020; Gourguillon et al., 2018; Hajati et al., 2016; Jia et al., 2014; and Tokuhara and Mii et al., 2001), exhibit variations of SD with respect to the mean value between 3 and 102 %, and as preliminary process it is not considered to compromising the reliability on the data reported and the cell culture behavior.

  • Liu et al., 2021. DOI: https://doi.org/10.1007/s11240-021-02139-7
  • El-Shafey et al., 2019. DOI: 10.21608/ejbo.2019.4873.1202
  • Jamil et al.,2018. DOI: 1007/s13205-018-1336-6
  • Gao et al., 2011. DOI: 10.1007/s11033-010-0245-5
  • Siwach et al., 2011. DOI: 10.5897/AJB10.2119
  • Deng et al., 2020. DOI: 10.3390/plants9111436
  • Gourguillon et al., 2018. DOI: //dx.doi.org/10.1016/j.scienta.2017.08.001
  • Hajati et al., 2016. DOI: 10.1007/s11627-016-9773-6
  • Jia et al., 2014. DOI: http://dx.doi.org/10.1016/j.scienta.2014.07.018
  • Tokuhara and Mii et al., 200. DOI:10.1079/IVP2001201

Q9. The NP extract exhibits the highest antioxidant activity, and its phytochemical profile is also different from other cell culture extracts. In Fig. 3, the NP chromatogram showed three major peaks on about 16 min, 18.5 min, and 24.5 min. Whether major peaks are unique components in the wild plant? If these major peaks are abundant, they may contribute greatly to the antioxidant activities. Therefore, it should be important to identify these compounds.

R. All the extracts were analyzed by HPLC (data not included) and compared against the elution time of 13 standard compounds: 3,4- dihydroxybenzoic acid; caffeic acid; catechin; epicatechin; gallic acid; taxifolin; naringenin; myricetin; kaempferol; isorhamnetin; p-coumaric acid; rutin; quercetin, and 3- epilupeol the last reported as the major compound with relevant biological activity for this plant, nevertheless, only six of these compounds were identified, as reported in the manuscript. We agree with Reviewer 1 that the identification of the major peaks on the chromatograms is relevant in their association with a specific contribution to the antioxidant activity of the extracts. Until now, these extracts are under consideration for HPLC-MS-MS analysis after fractionation and purification processes, because these peaks have not been reported for this plant, but this work is being considered for a different type of contribution, mainly centered on their purification and association with other specific biological activities. We provide some additional information regarding the results obtained until this moment for your consult, but they are still in processing. 

Q10.  A similar question is the same as question 4. There are two major peaks on about 21 min, and 27 min, and these peaks are unique shown in the chromatograms of CC and CSC. Therefore, it should be also important to identify these compounds.

R. We agree with the reviewer regarding the process for the identification of unknown compounds in the extracts, but as mentioned above, the process for their identification and their relationship to specific biological activities is currently on its way, being out of the scope of this manuscript.

Q11. The title of x-axis in the figures needs to be revise to Trolox equivalent concentrations.

R. We thank Reviewer 1 for the comment. To avoid the misunderstanding in the x-axis label, in lines 231-233, it was stated that Figure 4 shows the effect of the amount of dried extract per milliliter on the percentage of inhibition of the radicals DPPH, ABTS, and lipid peroxidation, and their comparison against different concentrations of Trolox.

Q12. Please explain which data is used for correlation analysis.

R. Data obtained for total phenolic compounds (TPC), flavonoids (TFC), and the IC50 values for ABTS and DPPH radical scavenging activity, TBARS inhibition of lipid peroxidation, were used for the analysis or Pearson correlation, to provide an explanation on the contribution of TPC and TFC to the antioxidant activity, rather than attributing this activity to specific compounds. This information was incorporated into the manuscript.

Reviewer 2 Report

This manuscript aims to provide information on phenolic compounds from wild plants and in vitro cultures of Ageratina pichichensis and the evaluation of their antioxidant activity. The presented subject can attract wide attention from researchers, because in vitro plant cell culture is an ecological alternative for increasing the production of bioactive compounds as systems that ensure the continuous production of bioactive compounds.

I recommend the publication. 

Author Response

The authors thank the comments and suggestions made by the reviewers, they help us to improve our manuscript, the response to each query is given below. All the changes were marked in red in the revised version of the manuscript.

Reviewer 2

Q1. This manuscript aims to provide information on phenolic compounds from wild plants and in vitro cultures of Ageratina pichichensis and the evaluation of their antioxidant activity. The presented subject can attract wide attention from researchers, because in vitro plant cell culture is an ecological alternative for increasing the production of bioactive compounds as systems that ensure the continuous production of bioactive compounds. I recommend the publication. 

R. We thank Reviewer 2 for the feedback on the manuscript.

Reviewer 3 Report

The Ageratina pichichensis is an interesting species that is worth of investigation, however the manuscript needs improvement as follow:

The photographs require the scale bars.

All the abbreviations on Figure 3 shall be explained in the figure caption.

The highest peaks presented on the chromatograms (Figure 3) shall be identified.

The first sentence in the subsection “2.4. Antioxidant activity” shouldn’t start with the expression “Because” – please change it.

The differences between results obtained from three assays (DPPH, ABTS, TBARS) need to be discussed in therms of the individual phenolic compounds that were detected in extracts from callus and suspension cultures as well as from in vitro plants and wild plants. Thus the conclusions also need improvement.

Author Response

The authors thank the comments and suggestions made by the reviewers, they help us to improve our manuscript, the response to each query is given below. All the changes were marked in red in the revised version of the manuscript.

Reviewer 3

The Ageratina pichichensis is an interesting species that is worth of investigation, however the manuscript needs improvement as follow:

Q1. The photographs require the scale bars.

R. The scale bars were included in Figures 1b and 1c, where the germinated plant in vitro and callus cell culture are shown, as requested by Reviewer 3.

Q2. All the abbreviations on Figure 3 shall be explained in the figure caption.

R. Abbreviations for the type of plant and cell culture as well as the chemical compounds identified in the extracts were provided in the caption of Figure 3.

Q3. The highest peaks presented on the chromatograms (Figure 3) shall be identified.

R. All the extracts were analyzed by HPLC and compared against the elution time of 13 standard compounds: 3,4- dihydroxybenzoic acid; caffeic acid; catechin; epicatechin; gallic acid; taxifolin; naringenin; myricetin; kaempferol; isorhamnetin; p-coumaric acid; rutin; quercetin, and 3- epilupeol, the last reported as the major compound with relevant biological activity for this plant, nevertheless, only six of these compounds were identified, as reported in the manuscript. We agree with Reviewer 3 that the identification of the major peaks on the chromatograms was not associated with any of these compounds remaining unidentified, nevertheless, these extracts are under consideration for HPLC-MS-MS analysis after fractionation and purification processes, these results are being considered for a different type of contribution, in which the association to other specific biological activities are monitored. We provide some additional information regarding the results obtained until this moment from this analysis.

Q4. The first sentence in the subsection “2.4. Antioxidant activity” shouldn’t start with the expression “Because” – please change it.

R. The wording for this sentence was revised.

Q5. The differences between results obtained from three assays (DPPH, ABTS, TBARS) need to be discussed in therms of the individual phenolic compounds that were detected in extracts from callus and suspension cultures as well as from in vitro plants and wild plants. Thus the conclusions also need improvement.

R. We thank Reviewer 2 for this comment. The discussion regarding the specific association of each compound to their contribution to the antioxidant activity is not suitable at this point. Different statistical tools were considered for giving a former approach to this correlation, but the results were not completely satisfactory, especially because there are several major and minor peaks that are not completely identified. It is noteworthy that the identification of phenolic compounds was done from crude extracts, where different types of compounds extractable in methanol solvent may not necessarily be of the type of phenolic compounds. Because the identification of unnamed compounds in this work is being done by fractionating and purifying them, for further analysis in HPLC-MS-MS, this analysis is beyond the scope of this manuscript.

Reviewer 4 Report

- Please write in full DPPH, ABTS, and TBARS assays at the introduction topic, which is the first mention in the Manuscript;

- Please define "IP" in topic 2.1 of results and discussion in the first mention;

- The PGR concentrations should be expressed only in micromolar. Please convert it along the manuscript;

- The expression "callus disintegration" is not suitable for describing the process the authors want to indicate. Instead of that expression, I suggest using "callus disaggregation". Please modify throughout the manuscript;

- Captions for figures and tables must be self-explanatory. Please include the meaning of the abbreviations in the legends and other information that is important for the interpretation of the results;

- Authors should rewrite the sentence "These results indicate that in vitro techniques promote and increase the production of phenolic compounds", including that in vitro techniques can promote phenolic compounds synthesis for the target species of the study, without generalizing;

- Topic 3.2.1: Please replace the term "planted" with "inoculated".

- The conclusions of the study can be improved. Authors should make this topic concise and to the point, keeping only the most relevant information. Conclusions should not be a summary of the main results, but rather a topic in which the authors succinctly present the main differentials of the study and what it was possible to advance in knowledge.

Author Response

Response to reviewers

The authors thank the comments and suggestions made by the reviewers, they help us to improve our manuscript, the response to each query is given below. All the changes were marked in red in the revised version of the manuscript.

Reviewer 4

Q1. Please write in full DPPH, ABTS, and TBARS assays at the introduction topic, which is the first mention in the Manuscript.

R. The complete names of the radical compounds were defined in the manuscript in the Introduction section.

Q2. Please define "IP" in topic 2.1 of results and discussion in the first mention.

R. The abbreviation “IP” was already defined in the Introduction section (Line 69).

Q3. The PGR concentrations should be expressed only in micromolar. Please convert it along the manuscript.

R. The concentrations for plant growth regulators reported in Table 1 and section 3.2.2 were expressed in micromolar, as requested by the reviewer.

Q4. The expression "callus disintegration" is not suitable for describing the process the authors want to indicate. Instead of that expression, I suggest using "callus disaggregation". Please modify throughout the manuscript.

R. We agree with reviewer 3, now it can be read as callus disaggregation.

Q5. Captions for figures and tables must be self-explanatory. Please include the meaning of the abbreviations in the legends and other information that is important for the interpretation of the results.

R. All the captions were revised, and all the abbreviations used in the figures were already defined for a better understanding.

Q6. Authors should rewrite the sentence "These results indicate that in vitro techniques promote and increase the production of phenolic compounds", including that in vitro techniques can promote phenolic compounds synthesis for the target species of the study, without generalizing.

R. The wording of this sentence was revised and modified as suggested by the reviewer.

Q7.- Topic 3.2.1: Please replace the term "planted" with "inoculated".

R. The wording was revised, and the term “planted” was changed by “inoculated”, as suggested by the reviewer.

Q8. The conclusions of the study can be improved. Authors should make this topic concise and to the point, keeping only the most relevant information. Conclusions should not be a summary of the main results, but rather a topic in which the authors succinctly present the main differentials of the study and what it was possible to advance in knowledge.

R. The Conclusions section was revised and improved, and unnecessary information was deleted to make it clearer and more concise.

Round 2

Reviewer 1 Report

The revised manuscript is fine. I agree to accept for publication.

Author Response

The authors thank to the comments and suggestions made by the Reviewer 1, certainly they help us to improve our work.